# Maltose and Maltotriose Transporters in Brewer’s *Saccharomyces* Yeasts: Polymorphic and Key Residues in Their Activity

**DOI:** 10.3390/ijms26135943

**Published:** 2025-06-20

**Authors:** Oscar A. Faz-Cortez, Jorge H. García-García, Ana K. Carrizales-Sánchez, Hector M. Fonseca-Peralta, Jessica G. Herrera-Gamboa, Esmeralda R. Perez-Ortega, César I. Hernández-Vásquez, Benito Pereyra-Alférez

**Affiliations:** 1Facultad de Ciencias Biológicas, Instituto de Biotecnología, Universidad Autónoma de Nuevo León, Cd. Universitaria, San Nicolás de los Garza 66455, Nuevo León, Mexico; oscar.fazcr@uanl.edu.mx (O.A.F.-C.); jorge.garciagr@uanl.edu.mx (J.H.G.-G.); cesar.hernandezvsqz@uanl.edu.mx (C.I.H.-V.); 2Department of Research and Development, Cervecería Cuauhtémoc Moctezuma, Monterrey 64410, Nuevo León, Mexico; ana.carrizales1@heineken.com (A.K.C.-S.); hector.fonseca@heineken.com (H.M.F.-P.); jessica.herrera@heineken.com (J.G.H.-G.); esmeralda.perez@heineken.com (E.R.P.-O.)

**Keywords:** brewer’s *Saccharomyces* yeasts, α-glucoside transporters, key residues, polymorphisms, sugar consumption

## Abstract

Maltose and maltotriose are the most abundant sugars in brewing wort, and their transport represent a critical bottleneck in the fermentation process. This transport relies on specific transmembrane proteins; however, many yeast strains exhibit inefficient uptake of these sugars, particularly maltotriose. Addressing this limitation requires a comprehensive understanding of the factors influencing the transport of maltose and maltotriose. This review provides a detailed synthesis of the key characteristics and functions of the maltose and maltotriose transmembrane transporters identified in brewer’s *Saccharomyces* yeasts. Critical amino acid residues involved in transporter activity are also highlighted, and the impact of specific polymorphisms and sequence variations on sugar preference and uptake efficiency is examined. Furthermore, a thorough discussion of the most important reported residues is presented, underscoring the need to closely examine their amino acid composition to better understand transporter mechanisms, optimize their performance, and enhance fermentation outcomes.

## 1. Introduction

Increasing knowledge about sugar transport in *Saccharomyces cerevisiae* is worthwhile, not only because of its important role as a model eukaryotic organism for fundamental study, but also for its extensive applications in food fermentations and other industrial fields [1]. In the brewing industry, the efficient uptake of sugars by yeasts plays an important role. In addition to *S. cerevisiae*, *Saccharomyces pastorianus* strains are also widely used in the brewing industry. This yeast is an interspecies hybrid between *S. cerevisiae* and *Saccharomyces eubayanus*, exhibiting characteristics of both species [2,3]. These two species are the most important in brewing processes. *S. cerevisiae* strains produce ale-style beer, which ferments at temperatures from 16 °C to 24 °C, while *S. pastorianus* strains produce lager-style beer, fermenting at temperatures from 3.3 °C to 13 °C. Even though *S. cerevisiae* is widely used for bread, distilled beverages, and beer production, *S. pastorianus* represents approximately 90% of beer production [4,5].

An important issue in the brewing industry related to these yeasts is the consumption of maltose and maltotriose [6], which are the first and second most abundant sugars, respectively, in a traditional brewing wort [7]. Although there are multiple physical and chemical factors inherent to the fermentation conditions, the molecular and physiological characteristics of the yeast strain used are key determinants [8]. The bottleneck in the fermentation process is the introduction of sugars into the cytoplasm of the cell for subsequent metabolism [9]. This uptake occurs through transmembrane proteins in which key residues have been identified in their structure, which are determinants of their activity; furthermore, several of these residues are polymorphic, so a single amino acid change in their sequence can be the determining factor between efficient or inefficient fermentation [10,11,12,13,14].

This review recapitulates the most important findings related to α-glucoside transporters, offering a deeper comprehension of their molecular-level activity. Key and/or polymorphic residues identified as crucial for transporter activity are also recapitulated. To our best knowledge, this is the first review to delve into α-glucoside transporters in yeast and key residues in their structure, which play a fundamental role in the fermentation of maltose and maltotriose.

## 2. Sugar Consumption in Brewing Wort

For efficient fermentation, the fermentable sugars in the wort must be consumed by the yeast cells. While complete sugar uptake is often considered economically advantageous, certain brewing applications such as the production of low-alcoholic beverages may benefit from yeasts with lower attenuation [12].

The main fermentable sugars in a typical wort are maltose, which represents approximately 60%; followed by maltotriose at around 20%; with glucose, fructose, and sucrose together comprising the remaining 20% [7,15]. Glucose and fructose are the first sugars to be metabolized by the cell via passive facilitated diffusion, which does not require energy [16,17]. Once hexoses are consumed, maltose becomes the preferred sugar, followed by maltotriose. These sugars are transported through active transport and are hydrolyzed in the cytoplasm by α-glucosidase enzymes into glucose molecules, which can enter metabolic pathways, as shown in Figure 1 [18,19,20].

Several factors have been identified as influencing the fermentation and the uptake of maltose and maltotriose, including yeast strain, fermentation temperature, ethanol tolerance of the cells [21,22,23], nitrogen concentration and assimilation rate [24,25], wort gravity [26,27], ion concentration [28], wort sugar spectrum [29], pitching rate [30], wort oxygen levels [31], and wort pH [32]. However, the bottleneck in the consumption of these α-glucosides is their transport into the cell, which is orchestrated by specific transporters. These transporters are encoded by genes such as *MPH2*, *MPH3*, *AGT1*, *MTT1* (also called *MTY1*), *MALx1* (Figure 1), and other less-characterized genes [33,34,35,36,37].

At the genetic and molecular levels, several factors influence maltose and maltotriose consumption beyond the presence of these genes, including the copy number of transporter-encoding genes, variations in the promoter regions of these genes, and positive regulators of Mal transporters [13,38,39,40]. Another crucial factor in this transport is the polymorphism in key amino acids within the protein sequence of these transporters, as it has been reported that even a single change in the sequence can determine substrate preference and may even alter transporter activity entirely [10,11,12,14,41].

## 3. Common Structural Characteristics of α-Glucoside Transporters in Brewer’s Yeasts

All α-glucoside transporters in brewer’s yeasts belong to the Major Facilitator Superfamily (MFS), the largest and most diverse group of secondary transporters, characterized by high structural conservation despite notable sequence variations [42,43]. The canonical structure of these permeases includes 12 transmembrane α-helices (TMH), arranged in two 6-TMH bundles connected by a hydrophilic and flexible intracellular loop. Their N-terminal and C-terminal ends are exposed to the cytoplasm (Figure 2) [44,45,46,47].

Specific TMHs have been identified as key, such as TMH7 and TMH11, as they harbor important amino acids involved in substrate transport. Additionally, some of these residues are polymorphic, increasing the functional differences between permeases [10,11,12,13,14].

The MFS permeases can introduce substrates into the cell through a channel (Figure 3) via passive or active transport, with the classification depending on whether external energy is required. Regarding α-glucoside transporters in yeasts, these require energy to uptake sugars, specifically introducing the substrate via a proton symporter, which means they utilize the electrochemical proton gradient to transport sugars into the cell, even for downhill transport. Additionally, the cell extrudes K^+^ to maintain electroneutrality [47,48,49].

## 4. MAL *Loci*

### 4.1. General Aspects

The MAL *loci* consist of five unlinked *loci*: MAL1-4 and MAL6. The canonical structure of these *loci* includes three genes involved in sugar metabolism: the *MALx1* gene (*MALT*), encoding a transporter primarily for maltose and composed of 614 amino acids; the *MALx2* gene (*MALS*), responsible for a maltase enzyme with hydrolase activity that breaks α-glycosidic bonds within the cell; and the *MALx3* gene (*MALR*), acting as an activation factor that enables the expression of the other two genes in the *locus* in the presence of maltose (Figure 4). The letter “x” represents the number of any of the five *loci*. All *MALx1* proteins share at least 95% identity between them [33,51,52]. The structure described above is the canonical one for a MAL *locus*, but it has been found that its structure can vary, with more or fewer copies of any of the three genes, which can result in non-functional *loci* and phenotypic changes. This phenomenon can partly be attributed to the fact that they are located in subtelomeric regions with a high tendency for rearrangements [53,54,55,56]. In *S. pastorianus*, its aneuploid nature further enhances this event [13,57,58].

While the fermentation of maltose in *Saccharomyces* requires the presence of at least one of the five *loci* to synthesize the permease [59], the transporter activity of Malx1p has been a topic of controversy. It has been described that, in general, Malx1p is a high-affinity maltose transporter (*K_m_* of ~4 mM) [34,60]. However, in some cases, it has also been reported to exhibit low affinity for other sugars, such as turanose, trehalose, sucrose (in the case of Mal21p), and maltotriose (in the case of Mal31p and Mal61p) [37,38,61]. Nevertheless, despite these latter permeases being attributed with maltotriose transport, multiple studies have reported that none of them are capable of transporting maltotriose [12,34,37,49]. This discrepancy may be partly due to polymorphisms in the sequences used by some authors, which grant different affinities for other substrates, or impurities in the reagents used as substrates [58]. These factors are also why the K*_m_* values of each transporter will not be discussed in detail, as they tend to vary significantly between studies.

### 4.2. Regulation and Possible Evolutionary Origin of the MAL Loci

The expression of these genes is inactivated by glucose, and although it is not entirely clear how maltose is sensed to induce its expression, its presence has been reported as necessary for this process [62,63]. Furthermore, Mig1p, a zinc-finger class DNA-binding protein, is involved in the regulation mechanism, acting as a glucose-mediated repressor for these genes. Studies have shown that this protein is also capable of repressing the expression of *MALx3*, which encodes the MAL activator [64,65]. In the canonical structure of the MAL *loci*, there is an intergenic region between *MALx1* and *MALx2* (Figure 4), which is a divergent promoter of approximately 0.9 kb that allows for the transcription of the other two genes in the locus. This promoter consists of two TATA boxes, two Mig1p binding sites, three binding sites for the MAL activator, and a 147 bp repeat element [64,66,67,68].

The MAL1 *locus* is considered to be the progenitor locus of the other MAL *loci*, as it is found in all *S. cerevisiae* strains on chromosome VII [34]. This *locus* can harbor a *MAL11* gene, an *AGT1* gene or an *MTT1* gene [34,56].

Despite MAL *loci* being distributed across multiple chromosomes, they share high sequence homology, and their genes can complement each other in the event of a mutation in one of them. This suggests that during the evolution of this multigenic family, the MAL *loci* were translocated and dispersed to different chromosomes via a mechanism involving rearrangements of chromosome termini [53,69]. The origin of many *loci* in *S. pastorianus* remains ambiguous, as this species is a hybrid between *S. cerevisiae* and *S. eubayanus*. To clarify their origin, it is necessary to identify strains more closely related to the parental strains. Nevertheless, sequences have been found in the genome of *S. pastorianus* with high homology to MAL *loci* from its parent species [70,71]. These genes have been reported as being more highly expressed in lager strains compared to ale strains [58].

It has been discussed that these *loci* may have evolved to regulate their expression through tandem repeats of 147 bp within the divergent promoter that regulates *MALx1* and *MALx2* (Figure 4). These tandem repeats cause a decrease in the expression level of *MALx1* compared to *MALx2*, which would prevent the toxic effect of overexpressing genes encoding these permeases. The more tandem 147 bp repeats found in the promoter, the longer the distance between the binding sites for the MAL activator and the TATA box of *MALx1*. This structural change in the promoter results in a significant reduction in the expression of *MALx1* but only a slight reduction in the expression of *MALx2* [68].

Malx1p proteins, encoded by *MALx1* genes in MAL *loci* have also been described as having a post-translational regulation mechanism: catabolite inactivation, mediated by ubiquitination. Motifs have been found in Mal31p and Mal61p that act as target signals for their subsequent proteolysis. These sites are rich in proline, glutamate, serine, and threonine (PEST), and have been reported in the N-terminal cytoplasmic domain of the permease [35,72].

## 5. *AGT1* Gene

### 5.1. Main Elements

The *AGT1* gene is an allelic variant of *MAL11*, meaning that it occupies the same chromosomal position within the *MAL1 locus* on the right arm of chromosome XII in *S. cerevisiae* (Figure 5) [34,56]. While *AGT1* is often referred to as *MAL11* by some authors and databases, it is important to distinguish between the two. The originally described *MAL11* gene, also located at the *MAL1 locus*, encodes a transporter that shares high sequence identity (95%) with other Malx1 proteins. In contrast, *AGT1* differs significantly in sequence, showing approximately 57% amino acid identity with Malx1 proteins [34]. Therefore, although *AGT1* and *MAL11* are allelic and sometimes used interchangeably in naming, they encode distinct transporters.

The *AGT1* gene encodes a broad-spectrum α-glucoside permease protein capable of transporting maltose and maltotriose in *S. cerevisiae*, as well as turanose, isomaltose, α-methylglucoside, palatinose, trehalose, turanose, and melezitose [34]. It was the first permease reported to be involved in the transport of other α-glucosides. Nearly all brewing strains are known to have this gene, and it has been identified as a key player in fermentations with *S. cerevisiae*, being considered the primary transporter of maltotriose in this species [34,37,49,57].

Due to the hybrid nature of *S. pastorianus*, a species formed between *S. cerevisiae* and *S. eubayanus*, this gene is present in both versions in *S. pastorianus*, with each originating from one of its parents, sharing 85% identity between both sequences [73,74]. However, in *S. pastorianus*, the gene originating from *S. cerevisiae* (*ScAGT1*) does not encode a functional permease, as it contains a nucleotide insertion that results in a premature stop codon, leading to a truncated, non-functional 394-amino-acid polypeptide. In contrast, the gene proposed to be donated by *S. eubayanus* (*lgAGT1*, also known as *SbAGT1* or *SeAGT1*) encodes a complete 616-amino-acid protein [58,73].

### 5.2. Regulation and Possible Evolutionary Origin of AGT1

*AGT1* is a maltose-inducible gene, with its induction mediated by the MAL activator [34], although maltotriose has also been proposed as an inducer [37,49]. The regulatory mechanism is similar to that of *MALx1*, with a divergent promoter between the start codon of *AGT1* and *MAL12*, the gene encoding maltase. This intergenic region is 785 bp long, 90 bp shorter than the intergenic region between *MAL61* and *MAL62*. Additionally, the 315 bp region immediately upstream of *AGT1* shows low identity with the comparable region in *MAL61*, but after 316 bp, the remaining 469 bp are nearly identical, with only a single nucleotide change across the sequence, where the MAL activator binding sites are located [34]. The elements in the promoter of the reference strain S288C have been reported as one Mig1p binding site and three MAL activator binding sites (Figure 5); however, the number and arrangement of these elements can vary depending on the species or strain [39].

More analyses are needed to characterize post-translational regulation by ubiquitination, but no PEST sequences were found in Agt1p as in Mal31p and Mal61p, suggesting potential differences in proteolysis mechanisms between these proteins and Agt1p.

This gene is strongly expressed in ale strains during growth in maltose, and weakly expressed in lager strains, while the opposite is observed with *MALx1* genes, which are more highly expressed in lager strains [58]. This difference in gene expression between species is mainly attributed to promoter structure differences, as ale strains’ sequences were found to contain an extra MAL activator binding element compared to the promoter of functional *AGT1* in lager strains [39].

It has been described that this gene is not part of a highly repeated family, unlike other *MALx1* genes, as it has only been found on chromosome VII in *S. cerevisiae*, while in *S. pastorianus*, the *SeAGT1* version was found on chromosome XV, originating from *S. eubayanus*. The origin of this allelic form of *MAL11* is attributed to a telomere translocation event at the MAL1 *locus*, such that the junction site between *AGT1* and *MAL21* would be at position 315–316 upstream of *AGT1* [34]. A similar translocation event is thought to have occurred in the evolution of the polygenic MAL family and the SUC *loci*. The formation of this allele is suggested to have involved two recombination events, one that translocated the *AGT1* gene into the subtelomeric region of chromosome VII, and another that occurred upstream of the *AGT1* and *MAL12* sequences, positioning the two genes next to one another while retaining MAL12’s MAL activator binding sites [34]. The precise origin of the *AGT1* sequence prior to these two recombination events remains unclear, though it has been speculated that it may be a chimeric gene, potentially sharing an evolutionary origin with the *MALx1* permeases [75,76].

## 6. *MPHx* Gene

### 6.1. General Features

The *MPH2* and *MPH3* genes, located on chromosomes IX and X, respectively, in *S. cerevisiae* (S288C strain), encode α-glucoside transporter proteins. These genes share 100% identity with each other, 75% identity with Mal61p and Mal31p, and 53% identity with the broad-spectrum transporter Agt1p [35].

The study of the substrates transported by *MPH2* and *MPH3* has been controversial and ambiguous. Initially, these permeases were characterized as transporters for maltose, maltotriose, α-methylglucoside, and turanose [35]. However, later studies contradicted these findings, suggesting that these transporters are incapable of transporting maltotriose or that their presence is not strongly associated with maltotriose consumption [49,54,77]. Other studies have argued that the reported ability of these permeases to transport maltotriose could be due to overestimation caused by substrate impurities, particularly with the use of [^14^C] maltotriose, which may have been contaminated (up to 16%) with [^14^C] maltose and [^14^C] glucose. Therefore, a more thorough characterization of these transporters is needed [49].

### 6.2. Regulation and Possible Evolutionary Origin of MPHx

In *S. cerevisiae*, the expression of *MPH2* and *MPH3* is induced by maltose and maltotriose and repressed by glucose. However, in lager strains, these genes are not expressed during growth in maltose or glucose, suggesting that their expression may be triggered by other conditions, such as the presence of maltotriose [58]. Their regulation is linked to the MAL activator encoded by *MALx3*, similar to the regulation of *AGT1* and *MALx1* genes [35].

The promoter regions of *MPH2* and *MPH3* share low sequence identity with the upstream regions of *MALx1* and *AGT1*, exhibiting only 45% and 43% identity, respectively. Additionally, only one MAL activator binding site is located 527 bp upstream of the start codon of *MPHx*, whereas *MALx1* and *AGT1* typically have three MAL activator binding sites. This suggests that the regulation of *MPHx* genes is not tied to a divergent promoter, as seen in the other reported permeases. Furthermore, only one binding site for the glucose repressor Mig1p was found 233 bp upstream of the *MPHx* sequences, compared to two Mig1p binding sites in the intergenic regions (promoters) *MALx1* [34,35,64,65].

These genes were not detected in any of the laboratory strains tested during their characterization [35], suggesting that the *MPH2* and *MPH3* sequences arose from a recent event. The hypothesis is that these genes were the result of a duplication event, possibly as a response to selective pressure, which could have been driven by other hexose transporters (HXTs) located in the same duplicated region. This process might have been amplified by the fact that subtelomeric regions, where these genes are located, are more prone to rearrangements [35,54,57].

## 7. *MTT1*/*MTY1* Gene

### 7.1. Overall Characteristics

The *MTT1* and *MTY1* genes were identified in the same year in lager strains by two unrelated research groups [36,37]. These genes are regularly referred to and treated as if they were the same gene; however, their open reading frames (ORFs) have 98% identity between them. The amino acid differences between them are distributed across cytoplasmic and extracellular loops, as well as within TMHs, including TMH5 and TMH6. Nevertheless, since most authors refer to them as one, the name *MTT1* will be used throughout this review to refer to *MTT1*/*MTY1* as a single gene.

Its ORF shows 74% identity with *MPHx*, 62% with *AGT1*, and 91% with *MAL61* and *MAL31*. Differences in amino acids between this protein and the Malx1p proteins have been identified, distributed along all the transmembrane regions; however, the 60 amino acids at the N- and C-terminal ends are practically identical. This gene encodes an α-glucoside transporter with 615 amino acids and is the only one reported to have a higher affinity for maltotriose than for maltose, with a *K_m_* of 16 to 27 mM and a *K_m_* of 61 to 88 mM, respectively. It also has the capacity to transport turanose and trehalose but not sucrose or α-methylglucoside [36,37]. Additionally, an important characteristic of this transporter is its functionality at low temperatures (≤10 °C), outperforming Agt1p in this respect. This factor could be significant in the ability of lager strains to ferment efficiently at lower temperatures compared to ale strains [15,78].

### 7.2. Regulation and Possible Evolutionary Origin of MTT1

Like other transporters, this one is repressed by glucose, and it has been identified that upstream of this gene is the α-glucosidase coding gene, and downstream is the gene encoding the transcriptional activator MAL. Recent research has reported the *MTT1* gene in the MAL1 *locus* in *S. cerevisiae*, in the same position where *AGT1* and *MAL11* can also be present [56]. Thus, this gene follows the canonical organizational structure of the MAL *loci* [36,37].

In lager strains, two versions of this gene have been reported, differing in size due to variations in their promoter regions. One measures 2.4 kb, while the other extends to 2.7 kb, and given that lager strains possess multiple chromosomes, it is likely that both versions coexist in multiple copies. This size difference results from the insertion of two tandem repeats of 147 bp within the promoter, adding a total of 294 bp to the longer variant. Interestingly, the shorter version of the gene is expressed at higher levels than the longer one [40]. A similar phenomenon had been reported previously in the MAL *loci* [68] (see above).

Two binding sites for the MAL activator have been found in both the short and long versions of this gene, as well as a Mig1p binding site, which partially overlaps with one of the MAL activator binding sites. However, in the long sequence, the distance between these binding sites is 294 bp further from the transcription initiation site compared to the short version due to the insertion of two 147 bp tandem repeats. This increase in distance between these regions may influence the difference in expression between these two versions of *MTT1* [40,68].

Several authors have discussed that this gene may have a chimeric nature, meaning it arose from a recombination event between two genes from *S. cerevisiae* and *S. eubayanus*, the progenitors of *S. pastorianus*. Given the high identity between *MTT1* and *MALx1*, it has been suggested that *MALx1* from *S. cerevisiae* gave rise to *MTT1*, while phylogenetic analyses propose *MALT3*, also an α-glucoside transporter, as the progenitor from *S. eubayanus* [37,75,79,80]. It has also been proposed that this gene later accumulated mutations during its evolutionary process following the hybridization event.

This gene exhibits a chimeric structure similar to *SeMALT413*, which encodes a permease capable of transporting maltotriose. *SeMALT413* was obtained through laboratory evolution of an *S. eubayanus* strain, resulting from repeated gene introgressions by non-reciprocal translocation of at least three *SeMALT* genes, a similar event that may have given rise to *MTT1* [76]. Another factor proposed to have influenced the evolution of this gene is the loss of functional *AGT1* genes in lager strains, which would provide more space in the plasma membrane for improved *MTT1* performance [78].

However, the presence of the *MTT1* gene has been reported in an *S. cerevisiae* strain through probe hybridization [15] and, more recently, in the MAL1 *locus* of *S. cerevisiae* [56]. These reports cast doubt on the hypotheses regarding the involvement of *S. eubayanus* in its origin. Nevertheless, it is important to note that other authors have reported a gene with 99% nucleotide identity to *MTT1* and 97% amino acid homology in *S. cerevisiae* strains. This gene, named *ScMALT*#5 [12], was characterized and found to have different properties from *MTT1*, despite its high sequence identity.

The fact that *MTT1*-like genes have been reported in *S. cerevisiae* strains but exhibit functional differences from *MTT1* [12] raises the possibility that the gene identified by Magalhães et al. [15] and Weller et al. [56] as *MTT1* may not actually be this gene, but rather a highly similar sequence such as *ScMALT#5* or another homolog that the specificity of the technique used could not distinguish. This possibility supports the previously proposed hypothesis that genes from *S. cerevisiae* and *S. eubayanus* contributed to the emergence of *MTT1* [75,76]. It also suggests that *MTT1*-like genes, such as *ScMALT#5*, may have arisen through a different evolutionary event within *S. cerevisiae*. However, further investigation is necessary before this can be established as a solid hypothesis.

Supporting that, a recent work reported the amplification of a PCR product using primers designed for *MTT1* in a wild *S. cerevisiae* strain, which turned out to be a *MAL31*-like gene [14]. This finding highlights the need for more precise characterization of these transporters before definitively identifying them as *MTT1*. Furthermore, more robust evidence is required to better understand their possible evolutionary origins.

## 8. Other Reported Important Permeases

In addition to the transporters described previously, other permeases capable of transporting maltose or maltotriose of interest in a brewing context have been reported. The transporters mentioned in the following lines are less characterized than those already mentioned, but they are worth noting.

### 8.1. MALT434

The *MALT434* gene encodes a functional permease capable of transporting maltotriose. This gene resulted from a laboratory evolution process of a *S. eubayanus* strain and is characterized as a chimeric gene, likely the product of an ectopic gene conversion [75]. This means that in the recombination event that generated this new allele, two genes with homologous sequences that are not in the same *locus* were involved [81]. The *MALT434* gene emerged from a recombination between *MALT4* and *MALT3*, which are maltose transporters but incapable of transporting maltotriose in *S. eubayanus*. All changes occurred in an allele of *MALT4* within a single 230 bp region, and this event did not affect the sequence of the parental gene *MALT3*. The translocated region of *MALT3* encodes TMH11 and portions of TMH10 and TMH12. Additionally, it introduced 11 nonsynonymous mutations to the protein coding sequence of *MALT4* [82].

Predictions for the protein encoded by *MALT434* indicate a high similarity to the typical structure of MALT transporters; however, it was predicted that several of its helices would be shorter compared to those of its parental genes [75].

### 8.2. SeMALT413

In the same year that *MALT434* was reported, another gene with similar characteristics, *SeMALT413*, was also reported. The *SeMALT413* gene was obtained by subjecting an *S. eubayanus* strain to UV mutagenesis and laboratory evolution under brewing conditions. This gene would have originated from sequences of three genes encoding maltose transporters from the same yeast, *MALT3* and *MALT4*, just like *MALT434*, but now *MALT1* also participated in its formation. It was predicted that the *SeMALT413* protein would have one helix with residues exclusively from MalT4, four helices formed exclusively by residues from MalT1, and five helices formed exclusively by residues from MalT3, with the other two remaining helices composed of residues from more than one MalT. Additionally, no shortening of any helices was predicted, unlike what occurred with *MALT434*. The events involved in its formation would be multiple introgression events in the ORF of *MALT4* [76].

### 8.3. ScMALT#2 and ScMALT#5

The *ScMALT#2* and *ScMALT#5* genes were recently reported in a brewing strain of *S. cerevisiae*; these genes encode functional α-glucoside transporters of 614 and 615 amino acids, respectively, sharing 93% amino acid identity with each other. Furthermore, ScMalt#2p shares 92% and 95% identity with Mal61p and Mtt1p, respectively, while ScMalt#5p shares 89% and 97% identity with Mal61p and Mtt1p, respectively. The ScMalt#2p permease transports maltotriose, while ScMalt#5p exhibits bifunctionality, efficiently transporting both maltose and maltotriose. It is believed that the ability of these transporters to transport maltotriose may be due to the domestication of the strain under brewing wort, through recombination events among *MALx1* genes in events similar to those that gave rise to *MALT434* and *SeMALT413* [12].

## 9. Functional Divergence and Rapid Expansion of α-Glucoside Transporter Genes

Because all these genes are localized in subtelomeric regions, they are more prone to undergoing high rates of mitotic and meiotic recombination, duplications, and mutations, which lead to much faster evolution and functional divergence compared to those in non-subtelomeric regions [54,83,84,85,86,87,88].

The chimeric genes *MALT434* and *SeMALT413*, mentioned previously, are clear examples of neofunctionalization during laboratory evolution. The relatively easy generation of genes with functions absent in ancestral versions is mainly attributed to the high duplication rate within these subtelomeric gene families, which are considered the primary source of new genes [88,89]. This, combined with the fact that a single change in a key amino acid in the protein structure can impact substrate preference, provides cells with multiple tools to innovate their sugar uptake capabilities and to rapidly adapt to novel niches [12,14,54,75,76].

The extraordinary dynamism of subtelomeric gene families not only allows for cells to acquire different characteristics, as demonstrated by the examples of *MALT434* and *SeMALT413*, but it has also been suggested that this dynamism promoted the rapid expansion of these genes throughout yeast evolution. Phylogenetic analyses indicated that the common ancestor of yeast had only a few *MAL* genes, which were completely lost in some lineages and expanded in others [54]. An example is *Candida* spp., which colonize mammals and have lost all *MAL* genes, likely because these yeasts encounter simpler sugars such as those found in blood and the digestive tract [54]. On the other hand, yeasts adapted to brewing processes possess multiple copies of *MAL* genes, including some variations with enhanced characteristics for more effective uptake of complex sugars such as maltotriose [13].

Moreover, the existence of multiple subfamilies within these genes has been revealed. Interestingly, genes within a single subfamily represent not only orthologs but also paralogs, highlighting recent duplication events within the same species [54].

The evolution and diversification of transporters with specific functions has traditionally been regarded as an example of functional diversification, where novel substrate affinities are gained in a highly plastic manner. This perspective suggests that functional diversification occurs through neofunctionalization following the duplication of ancestral transporter genes [12,47,54]. However, recent studies propose an alternative explanation, suggesting that the evolution of this gene family may have primarily followed a different path. Specifically, duplications may have enabled the resolution of functional trade-offs present in ancestral promiscuous permeases, allowing for each duplicate to specialize in a subset of substrates, a process more consistent with specialization through subfunctionalization, rather than the acquisition of entirely new functions (neofunctionalization) [82]. α-Glucoside transporters likely evolved from promiscuous ancestral permeases, optimizing subfunctions following duplication events and leading to specialization for specific sugars [54,82,90,91]. The findings of Crandall et al. [82] and Hatanaka et al. [12] support this idea by suggesting that the acquisition of novel functions in high-specificity permeases is highly improbable and would require the simultaneous emergence of multiple interacting mutations. Nevertheless, further research is necessary to strengthen this hypothesis.

## 10. Key and Polymorphic Residues Involved in α-Glucoside Transporters Activity

Previously, the importance of gene duplications in acquiring proteins with distinct characteristics was highlighted, but it has also been described that certain amino acids are essential for protein activity, and some of these have been identified as polymorphic residues. These residues can determine substrate preference, especially if they are located near the substrate transport channel. Below is a table summarizing the most important residues reported to date and the reasons they are considered crucial or potentially crucial for protein activity (Table 1).

### 10.1. Polymorphic Residues in α-Glucoside Transporters

There is a very high number of polymorphisms among all permeases of this type. However, the present review focuses on and report only those that have been previously reported in research and have been discussed as potentially playing an important role in transporter activity due to their proximity to the substrate transport channel or interactions with residues in key helices.

The residues around the channel could directly influence substrate affinity by making direct contact with the sugar through electrostatic interactions or indirectly by interacting with amino acids around it, altering the stability and structure of the protein [12,82,95]. Although many residues in the 12 TMHs are near the channel or important residues for substrate transport, TMHs of greater importance have been identified, such as TMH7 and TMH11, which face each other. These helices not only contain amino acids that have been reported as crucial for substrate preference but have also been identified as highly polymorphic regions at key residues among permeases like Agt1p, Mtt1p, Mal31p, Mal61p, ScMalt#2p, and ScMalt#5p, as well as other hexose transporters in yeasts [12,13,14,41,82].

In TMH7, the residues at positions 371/372, 374/375, 375/376, 378/379, and 383/384 are polymorphic among Mal31p sequences, which in turn differ in those same amino acids in Mtt1p, Mal61p, ScMalt#2p, and ScMalt#5p (Figure 6), with those permeases capable of transporting maltotriose having V-T-T-T-N at those positions. It has been hypothesized that this set of polymorphic residues (V371/372, T374/375, T375/376, T378/379, and N383/384) could be an evolutionary product, as the strains are under constant pressure from high sugar concentrations, since this set of residues has been reported in sequences of brewing strains [12,14]. In fact, a study comparing two brewing strains found that the one with this set of polymorphic residues in that sequence in some copies of its Mal31p was significantly more efficient at consuming maltose and maltotriose than the one that did not have them [13]. Although multiple factors influence the consumption of these sugars, key polymorphic residues in important TMHs have been proposed to have a strong impact on transport activity [10,11,12,14].

The polymorphic residues of TMH11 have been identified as key in substrate preference in Mtt1p, Mal61p, ScMalt#2p, and ScMalt#5p [12]. Although other polymorphic residues located in other helices, such as TMH1, TMH2, or TMH12, have not been experimentally demonstrated to intervene in the transport of these sugars, it is possible that they do play an important role in their transport or substrate affinity, as a single change can alter their flexibility and stability [14,41]. Therefore, it is important to continue more detailed studies of them.

It has even been recommended to sequence all these genes from each strain intended for fermentative purposes, as it is clear that a single change in its amino acid sequence can drastically affect substrate affinity [58]. Furthermore, key polymorphic residues could be used as molecular markers to obtain information on the potential fermentative capacity that a strain might have or even to gain more detailed knowledge of its genetic tools for evolving toward better performance [14].

### 10.2. Key Residues Involved in Sugar Transport in α-Glucoside Transporters

Charged residues generally play an important role in proton and sugar binding in H^+^-symporters. Mutations in key sugar-binding residues in MFS have shown that they can completely abolish transport, while key residues in proton translocation may still allow transport, although their activity and affinity decrease [96,97].

Trichez et al. [11] identified four charged residues that form the substrate transport channel in Agt1p: E120 (TMH1), D123 (TMH1), E167 (TMH2), and R504 (TMH11) (Figure 7B). These residues are highly conserved across all characterized α-glucoside transporters in yeasts and fungi. Substituting each of these residues individually with small nonpolar amino acids drastically reduces Agt1p’s transport efficiency for maltose, trehalose, sucrose, and α-methylglucoside. In the case of maltotriose, its uptake is also impaired, particularly with the R504 substitution, which completely abolishes transport. E120, D123, and E167 have been proposed to play a role in proton translocation in Agt1p [92]. While E167 appears to have a more direct role in proton coupling, D123 is also a conserved residue in other proton-translocating permeases (or is replaced by another acidic residue at the same position), and its substitution disrupts normal transporter activity [92,98,99]. In addition to those residues, Q225 has been suggested as a potential substrate binding site, and although conclusive assays confirming its role as a key residue are lacking, mutations at this position would likely affect Agt1p’s transport activity [11].

Variants of Agt1p have been reported, and important amino acids involved in its transport activity have been identified. A study comparing the Agt1p sequences of two *S. cerevisiae* strains found that one strain (WH314) carries a 23-amino-acid truncation at the C-terminus, resulting in significantly reduced transport activity compared to the Agt1p from strain WH310. However, despite this truncation, it was reported that the difference in maltotriose transport is not due to the truncation itself but rather to three specific amino acids present in the Agt1p of WH310 but absent in WH314. Two of these residues, T505 (TMH11) and S557 (cytoplasm) (Figure 7B), were identified as critical for maltotriose transport in WH310, as their absence completely abolishes the permease’s ability to transport this substrate. The third residue, V549 (TMH12) (Figure 7B), while not essential, plays a supporting role in transport efficiency [10,100].

Although the degradation mechanism of Agt1p may differ from that reported for Malx1p due to the absence of PEST sequences [34,35], S557 is believed to be crucial for Agt1p degradation or other key intracellular processes. Despite being located in an intracellular loop (Figure 7B), its presence is essential for maltotriose uptake in WH310. In contrast, WH314 carries a T557 substitution, which could make its Agt1p more prone to degradation, potentially functioning as a phosphorylation site. Meanwhile, T505 likely plays a direct role in maltotriose interaction, as it is exposed to the transport substrate channel, whereas V549 appears to contribute mainly to protein folding and stability [10,100].

As mentioned before, Agt1p transports a wide range of α-glucosides, including trehalose. This review focuses on maltose and maltotriose, as they are the most abundant sugars in a typical brewing wort. However, it is worth mentioning that a recent study has characterized several residues important for trehalose transport, such as Q156, L164, Q256, E395, R396, Y507, Q137, T230, and N514 [101]. These findings provide crucial insights into the transport of this α-glucoside, which plays an important role in cellular stress resistance [102].

Hatanaka et al. [12] identified two genes in a brewing *S. cerevisiae* strain whose encoded proteins share high amino acid identity with Mtt1p. These transporters, named ScMalt#2p and ScMalt#5p, facilitate maltotriose transport, with ScMalt#5p also exhibiting bifunctionality for both maltose and maltotriose. The same study determined that substrate specificity in ScMal61p, Mtt1p, ScMalt#2p, and ScMalt#5p is influenced by polymorphic residues in TMH7 and TMH11 of these permeases (Figure 7A) [12].

In TMH7, two important residues were reported: ScMalt#2p (T378 and N383), ScMalt#5p (T379 and N384), Mtt1p (T379 and N384), and Mal61p (A378 and Y383). These residues are exposed to the substrate transport channel (Figure 7A), and their side chains point toward this channel. Growth assays confirmed that Thr and Asn in TMH7 are key for maltotriose transport in ScMalt#5p, ScMalt#2p, and Mtt1p, as substituting just these two amino acids with Ala and Tyr (from Mal61p) causes them to lose their ability to transport maltotriose entirely, while conferring maltose transport capability to Mtt1p and ScMalt#2p, which previously lacked it (the version of Mtt1p used did not transport it). These two substitutions not only strip ScMalt#5p of its ability to transport maltotriose but also significantly increase its maltose consumption rate. This, in turn, provides strong insights into the effect that A378 and Y383 have on maltose preference in Mal61p, demonstrating that only these two substitutions are enough to define the preference between maltose and maltotriose in these transporters, which in turn reflects their ancestral preference for maltose. Y383 is a bulkier amino acid than Asn, found in maltotriose transporters. This difference in size might spatially restrict maltotriose (three glucose units) transport in Mal61p but allow for maltose (two glucose units) transport [12].

Seven polymorphic amino acids in TMH11 are important for substrate recognition. Of these, only residue 506 is positioned near the transport channel, with its side chain oriented toward it. The remaining six residues are either distant from the sugar-binding site or have side chains that do not face the channel. The TMH11 sequences of ScMalt#2p and Mtt1p are identical (T491/492, S493/494, A500/501, M503/504, A504/505, S505/506, I506/507), whereas ScMalt#5p (V492, T494, T501, V504, A505, S506, and I507) and Mal61p (T491, T493, A500, V503, I504, Q505, and V506) differ from each other.

Residues V492, T494, T501, and V504 have been identified as key for ScMalt#5p’s bifunctional capability, playing a crucial role in maltose assimilation. Introducing these residues into Mtt1p could potentially enable maltose transport. Similarly, residues A505, S506, and I507 have been identified as crucial for maltose transport in ScMalt#5p. On the other hand, residues T491/492, S493/494, A500/501, and M503/504 in ScMalt#2p and Mtt1p have a significant effect on maltotriose transport, and mutations introducing these residues into ScMalt#5p enhance its maltotriose transport efficiency [12].

Important residues have recently been identified in MalT434, a chimeric maltotriose transporter. As previously mentioned, this protein is the product of an ectopic recombination event between two MalT transporters, which occurred during the laboratory evolution of an *S. eubayanus* strain [75]. Further analysis of MalT434 and its parental genes provided insights into the acquisition of maltotriose transport capacity in this permease, revealing six crucial residues involved in this function. The novel ability to transport maltotriose requires a specific combination of three amino acids in TMH11: C505, T512, and either M503 or T508. Additionally, S379 (TMH7) and F468 (TMH10) are also essential (Figure 8). These conclusions were drawn from studies on MalT4, one of the parental genes of MalT434 [82].

The key amino acids in TMH7 and TMH11 facilitate a strong epistatic interaction between these two helices, an interaction previously reported as important for transporter function [12,14]. In *S. eubayanus* MalT transporters, this large interaction is primarily attributed to a single amino acid, S379, whose mutation completely abolished maltotriose transport. Moreover, all these residues are located near the transport channel, making direct interactions with the substrate likely. On the other hand, F468 is positioned farther from the transport channel, making direct substrate interaction unlikely. However, its importance may lie in subtle alterations to the overall protein conformation through interactions with other residues [82].

### 10.3. Key Residues Reported in Post-Translational Regulation of α-Glucoside Transporters

Glucose not only acts as a regulator of maltose transporters at the transcriptional level but also at the post-translational level, controlling the levels of the permease in the cell in a phosphorylation- and ubiquitination-dependent process, which requires the ubiquitin ligase Rsp5 and the ubiquitin hydrolase Doa4 [103,104,105,106]. Glucose and its non-metabolizable analog, 2-deoxy-D-glucose, are capable of inducing the phosphorylation and ubiquitination of permeases for their degradation in a process called catabolite inactivation, independent of the proteasome, in which prior internalization by endocytosis of the permease is required for its subsequent transport to vacuoles, where proteolysis takes place [104,105,106]. Phosphorylation is mediated by Protein Kinase A and Protein Kinase C [107], where putative phosphorylation sites have been identified in Mal61p—S295, S317, and S487—as targets for Protein Kinase C and T363 for Protein Kinase A. Mutants of Mal61p in which these amino acids were substituted showed a significant reduction in glucose-induced inactivation compared to wild-type Mal61p, especially the mutant where S295 was replaced, indicating a key role in catabolite inactivation [94].

Another sequence necessary for glucose-induced degradation has been reported, which is related to ubiquitin conjugation and phosphorylation in Mal61p and Mal31p. It is present in the N-terminal cytoplasmic domain, specifically between residues 49 to 78, corresponding to a putative PEST sequence (Figure 9) [33], named this way because it is rich in proline, glutamate, aspartate, and serine. Within this sequence, a dileucine motif (L69 and L70) (Figure 9) believed to be an important ubiquitin-binding site has been identified. Close to this putative PEST region within the N-terminal domain, other serine and threonine residues have been identified, which could also be phosphorylation sites [72,108].

The full set of events involved in glucose-induced inactivation of these permeases has not yet been fully defined or characterized, but it has been hypothesized that ubiquitination occurs after the phosphorylation of residues S295, S317, S487, and T363 (Figure 9) (not present in the N-terminal domain), an event that alters the local structure, making ubiquitination accessible, which is necessary for the proper internalization of the transporter for subsequent vacuolar degradation [94,105,106,109]. Studies suggest that the serine- and threonine-rich phosphorylatable regions, as well as the PEST region and the putative dileucine motif present in the N-terminal domain, do not participate in the internalization of permeases, but their importance in glucose-induced inactivation lies in targeting the internalized permease for its movement toward the vacuole and subsequent proteolysis. Wild-type Mal61p permeases containing these sequences are found in both the plasma membrane and vacuoles for degradation, whereas mutants in which these regions are altered are directed to another intracellular compartment often referred to as the prevacuolar or E-compartment, where they are not degraded [108].

In Mal21p, it has been reported the presence of amino acids that confer resistance to glucose-induced inactivation, significantly increasing its half-life compared to Mal61p and Mal31p in the presence of glucose and 2-deoxy-D-glucose, its non-metabolizable analog. These amino acids are G46 and H50, located in the N-terminal region of the protein (Figure 10). Their coexistence allows for Mal21p to be less susceptible to ubiquitination and internalization by endocytosis, increasing its permanence in the membrane compared to Mal61p and Mal31p. Comparing the sequences of Mal21p with Mal61p reveals that they differ in 10 amino acids distributed throughout the protein, two of which are G46 and H50, present only in Mal21p. It has been shown that Mal61p mutants in which the original amino acids (D46 and L50) were replaced with G46 and H50 acquire resistance to ubiquitination and degradation, similar to Mal21p. These findings are particularly interesting since, despite there being no difference in the number of lysines between Mal21p and Mal61p, Mal61p exhibits significantly more ubiquitination. The hypothesis regarding this resistance conferred by these two amino acids suggests that the N-terminal domain (rich in acidic residues) where D46 and L50 are found in Mal61p interacts with the long cytoplasmic domain (rich in basic residues) where S295 and T363 are located, as mentioned earlier. This stereostructural relationship between the two domains would influence the efficient ubiquitination of Mal61p, and it is possible that the lysine residues can only be ubiquitinated when the conformation of these domains is appropriate. This would explain why mutations in any of these four residues (S295, T363, D46, or L50) confer resistance to Mal61p against glucose-induced inactivation. In Mal21p, due to differences in residues in the N-terminal domain, there is likely a change in this stereostructural relationship between the two mentioned domains, making the lysines less susceptible to ubiquitination [93,94].

Post-translational regulation in other α-glucoside transporters has not yet been studied; however, it has been reported that Mal11p has a significantly longer half-life than Mal61p, despite having 99% identity [94]. On the other hand, no PEST sequences were found in Mphxp or Agt1p [35], suggesting a post-translational regulation system different from that reported for Mal31p and Mal61p, which could be related to the evolution of another proteolysis system, as Mphxp and Agt1p are believed to be more recent genes from an evolutionary perspective [34,35].

Recent research on the post-translational regulation of α-glucoside transporters remains limited. However, the crucial role this process plays in transporter activity has been highlighted here. Therefore, more recent investigations into this regulatory mechanism are necessary, particularly in newly identified transporters. Additionally, a more comprehensive characterization of post-translational regulation across all currently known important α-glucoside transporters is essential to deepen our understanding of their function and efficiency.

## 11. Conclusions

This review underscores the critical importance of examining the specific amino acid sequences of maltose and maltotriose transporters and highlights how even single amino acid variations can dramatically influence their activity. Detailed characterization of these sequences in strains intended for industrial use can offer valuable insights into their transport efficiency and provide markers for strain optimization.

Further research into the molecular mechanisms of maltose and maltotriose transporters is highly worthwhile—not only for improving their uptake efficiency in industries such as brewing but also for deepening our understanding of the intricate interactions between substrates and permeases. These transporters belong to a vast family of transmembrane proteins found across diverse organisms, many of which share fundamental structural and functional characteristics. Advancing knowledge in this area can therefore yield broader implications for fields beyond brewing, contributing to a more comprehensive understanding of membrane transport systems in general.

## Figures and Tables

**Figure 1 ijms-26-05943-f001:**
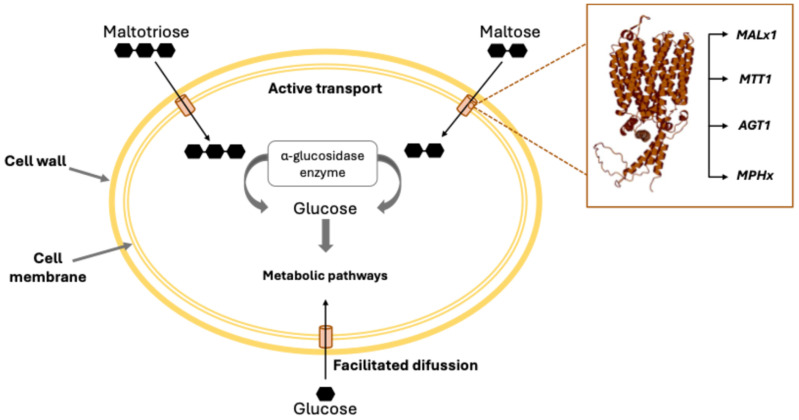
Schematic of the consumption of the three most important sugars in a typical brewing wort. Glucose is introduced via facilitated diffusion, while maltose and maltotriose are actively transported into the cytoplasm, where the enzyme α-glucosidase hydrolyzes them, releasing free glucose molecules [7].

**Figure 2 ijms-26-05943-f002:**
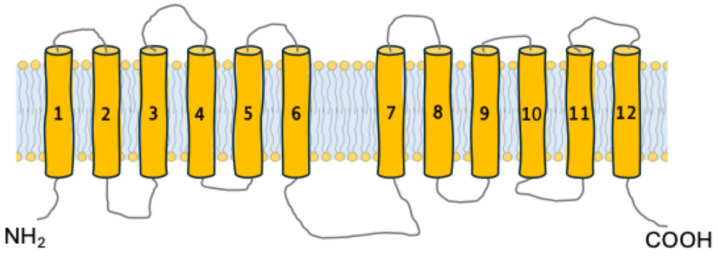
Topological structure of an α-glucoside transporter. The cylinders represent the 12 typical transmembrane helices [45,47].

**Figure 3 ijms-26-05943-f003:**
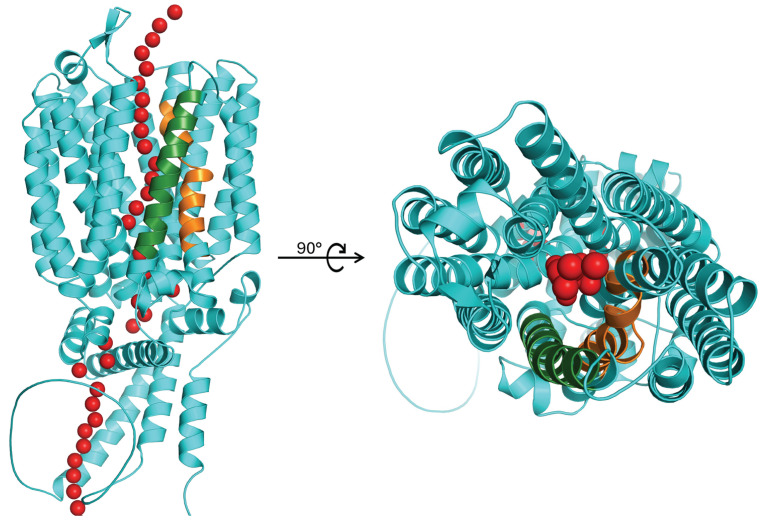
Representation of 3D structure of an α-glucoside transporter in yeast, along with the predicted substrate transport channel (represented by red spheres). TMH7 is highlighted in orange and TMH11 in green. Prediction of the substrate transport channel was performed using PoreWalker [50].

**Figure 4 ijms-26-05943-f004:**
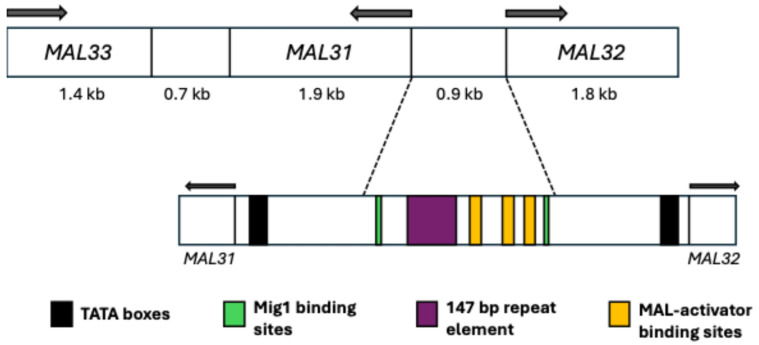
Canonical structure of the MAL3 *locus*. The intergenic region between *MAL31* and *MAL32* is highlighted, corresponding to the divergent promoter for both genes. Modified from Vidgren et al. [39].

**Figure 5 ijms-26-05943-f005:**
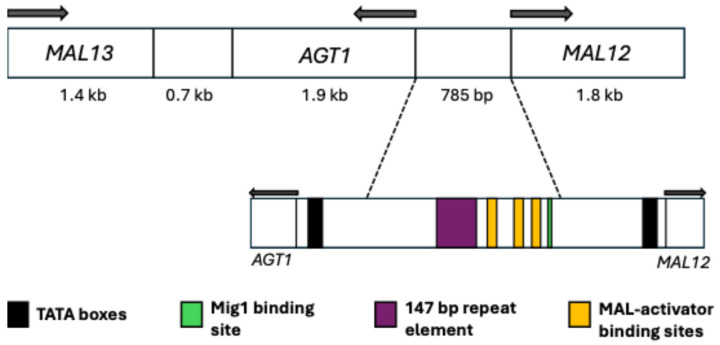
Canonical structure of the MAL1 *locus* in *S. cerevisiae* S288C harboring the *AGT1* gene. The intergenic region between *AGT1* and *MAL12* is highlighted, corresponding to the divergent promoter shared by both genes.

**Figure 6 ijms-26-05943-f006:**
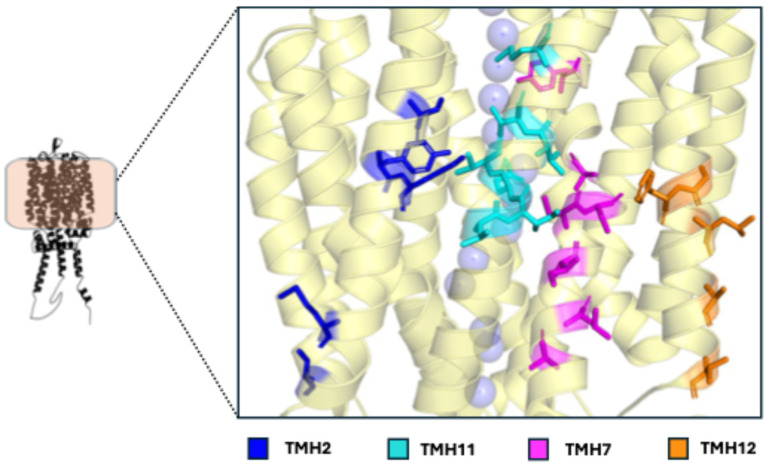
Polymorphic residues in Mal31p potentially important for its transport activity, highlighted in different colors corresponding to their TMH. The substrate transport channel is represented by purple spheres.

**Figure 7 ijms-26-05943-f007:**
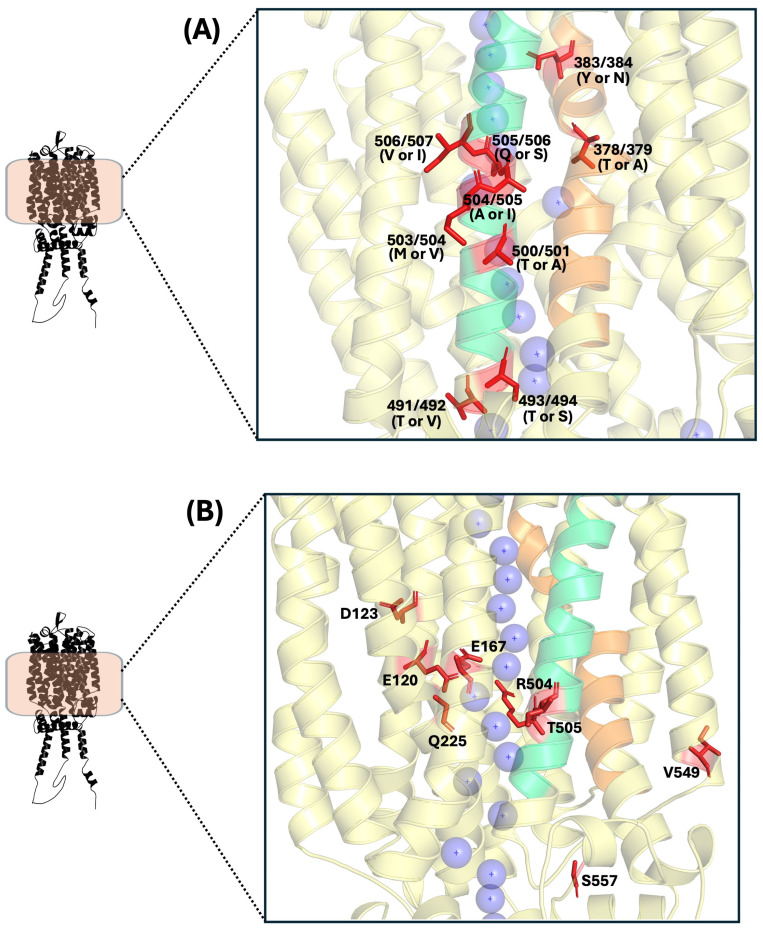
Key residues (in red) involved in maltose and maltotriose transport, with the substrate transport channel represented by purple spheres. TMH7 is shown in orange and TMH11 in green. (**A**) Key residues in ScMalt#5, ScMalt#2, and Mtt1p (using Mtt1p structure as a template). (**B**) Key residues in Agt1p.

**Figure 8 ijms-26-05943-f008:**
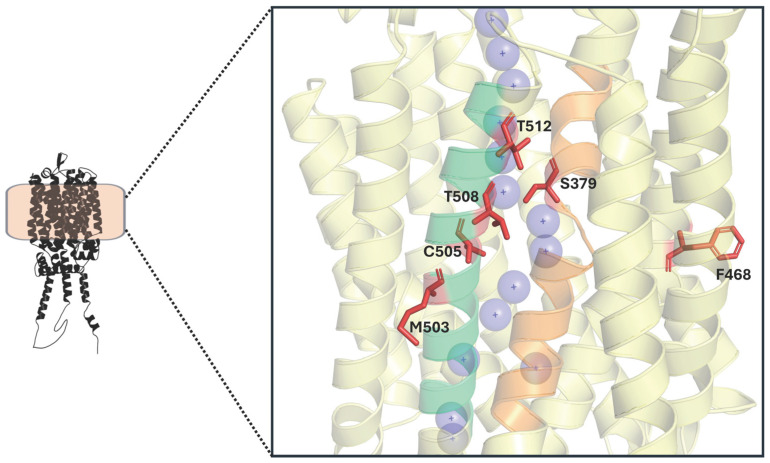
Key residues (in red) involved in maltotriose transport in MalT434. The substrate transport channel is represented by purple spheres. TMH7 is shown in orange and TMH11 in green.

**Figure 9 ijms-26-05943-f009:**
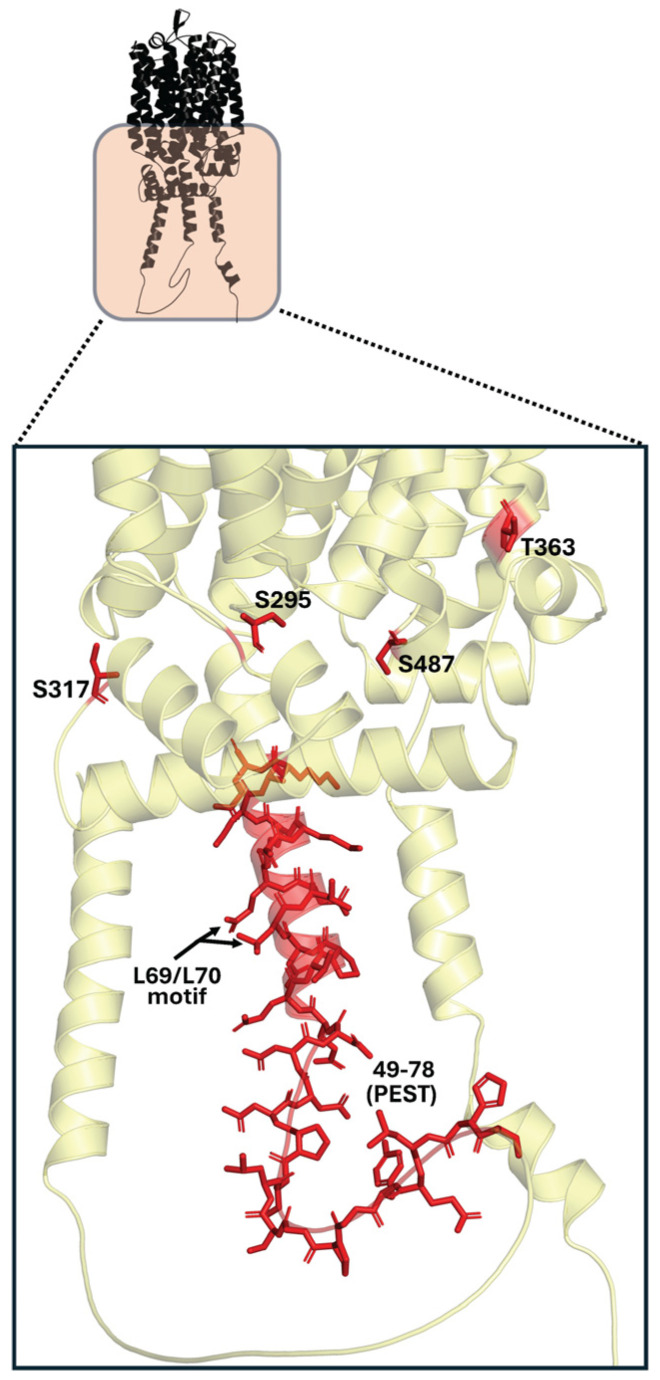
Key residues (in red) involved in phosphorylation and ubiquitination for glucose-induced inactivation in Mal61p and Mal31p.

**Figure 10 ijms-26-05943-f010:**
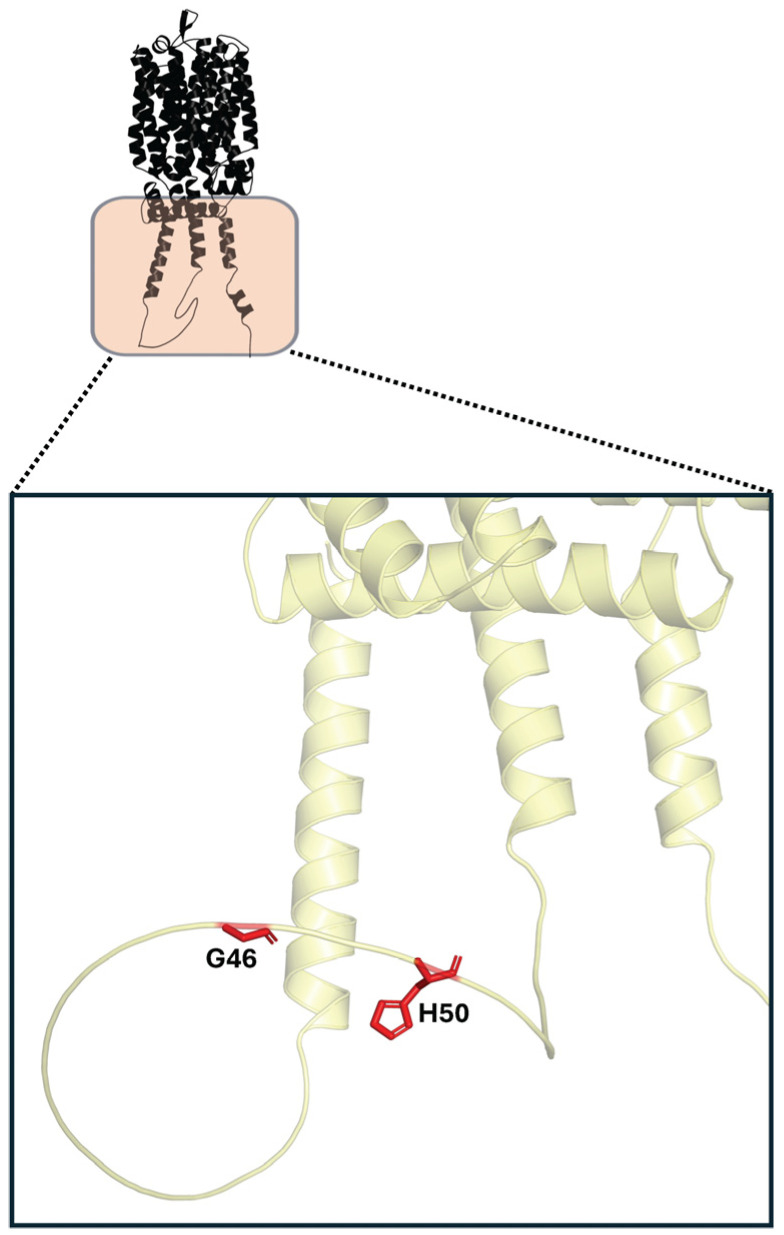
Key residues (in red) involved in the resistance of Mal21p to glucose-induced inactivation.

**Table 1 ijms-26-05943-t001:** Key residues reported involved in activity of α-glucoside transporters in *Saccharomyces* yeasts.

α-Glucoside Transporter	Key Residue (s)	Region	Importance	Proximity to Transport Channel
Agt1p [13]	S128	TMH1	Polymorphic residue	High
Agt1p [13]	L164	TMH2	Polymorphic residue	High
Mal31p [14]	V371	TMH7	Polymorphic residue	High
Mal31p [14]	C374	TMH7	Polymorphic residue	High
Mal31p [14]	S375	TMH7	Polymorphic residue	High
Mal31p [14]	A378	TMH7	Polymorphic residue	High
Mal31p [14]	Y383	TMH7	Polymorphic residue	High
Mal31p [13]	C154	TMH2	Polymorphic residue	High
Mal31p [13]	Y157	TMH2	Polymorphic residue	High
Mal31p [13]	M158	TMH2	Polymorphic residue	High
Mal31p [13]	M167	TMH2	Polymorphic residue	Medium
Mal31p [13]	S171	TMH2	Polymorphic residue	Medium
Mal31p [13]	V503	TMH11	Polymorphic residue	High
Mal31p [13]	I504	TMH11	Polymorphic residue	High
Mal31p [13]	Q505	TMH11	Polymorphic residue	High
Mal31p [13]	V506	TMH11	Polymorphic residue	High
Mal31p [13]	V508	TMH11	Polymorphic residue	High
Mal31p [13]	T509	TMH11	Polymorphic residue	High
Mal31p [13]	M513	TMH11	Polymorphic residue	High
Mal31p [13]	A366	TMH7	Polymorphic residue	High
Mal31p [13]	L368	TMH7	Polymorphic residue	High
Mal31p [13]	G533	TMH12	Polymorphic residue	Low
Mal31p [13]	F534	TMH12	Polymorphic residue	Low
Mal31p [13]	L536	TMH12	Polymorphic residue	Low
Mal31p [13]	A540	TMH12	Polymorphic residue	Low
Mal31p [13]	V544	TMH12	Polymorphic residue	Low
Agt1p [11]	R504	TMH11	α-glucoside transport	High
Agt1p [11]	Q225	TMH4	Substrate binding site	High
Agt1p [11,92]	E120	TMH1	α-glucoside transport, proton translocation	High
Agt1p [11,92]	D123	TMH1	α-glucoside transport, proton translocation	High
Agt1p [11,92]	E167	TMH2	α-glucoside transport, proton translocation	High
Agt1p [10]	V549	TMH12	Maltotriose transport, polymorphic residue	Low
Agt1p [10]	T505	TMH11	Maltotriose transport, polymorphic residue	High
Agt1p [10]	S557	Cytoplasm	Maltotriose transport, polymorphic residue	Medium
Mtt1p [12]	T379	TMH7	Maltotriose transport, polymorphic residue	High
Mtt1p [12]	N384	TMH7	Maltotriose transport, polymorphic residue	High
ScMalt#2p [12]	T378	TMH7	Maltotriose transport, polymorphic residue	High
ScMalt#2p [12]	N383	TMH7	Maltotriose transport, polymorphic residue	High
ScMalt#5p [12]	T379	TMH7	Maltotriose transport, polymorphic residue	High
ScMalt#5p [12]	N384	TMH7	Maltotriose transport, polymorphic residue	High
ScMalt#5p [12]	V492	TMH11	Maltose transport, polymorphic residue	High
ScMalt#5p [12]	T494	TMH11	Maltose transport, polymorphic residue	High
ScMalt#5p [12]	T501	TMH11	Maltose transport, polymorphic residue	High
ScMalt#5p [12]	V504	TMH11	Maltose transport, polymorphic residue	High
ScMalt#5p [12]	A505	TMH11	Maltose transport, polymorphic residue	High
ScMalt#5p [12]	S506	TMH11	Maltose transport, polymorphic residue	High
ScMalt#5p [12]	I507	TMH11	Maltose transport, polymorphic residue	High
MalT434 [82]	M503	TMH11	Maltotriose transport	High
MalT434 [82]	C505	TMH11	Maltotriose transport	High
MalT434 [82]	T508	TMH11	Maltotriose transport	High
MalT434 [82]	T512	TMH11	Maltotriose transport	High
MalT434 [82]	F468	TMH10	Maltotriose transport	Medium
MalT434 [82]	S379	TMH7	Maltotriose transport	High
Mal31p [35,72]	L69 and L70	Cytoplasm (N-terminal)	Important ubiquitin binding site	Low
Mal21p [93]	G46	Cytoplasm (N-terminal)	Resistance against degradation	Low
Mal21p [93]	H50	Cytoplasm (N-terminal)	Resistance against degradation	Low
Mal61p [94]	S295	Cytoplasm	Protein Kinase C site	Low
Mal61p [94]	S317	Cytoplasm	Protein Kinase C site	Low
Mal61p [94]	T363	Cytoplasm	Protein Kinase A site	Low
Mal61p [94]	S487	Cytoplasm	Protein Kinase C site	Low
Mal61p [72]	L69 and L70	Cytoplasm (N-terminal)	Important ubiquitin binding site	Low

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
