# Peer review of "Maltose and Maltotriose Transporters in Brewer’s Saccharomyces Yeasts: Polymorphic and Key Residues in Their Activity"

_ijms, 2025, doi:10.3390/ijms26135943_

Round 1

Reviewer 1 Report

Comments and Suggestions for Authors

In this manuscript, the authors review the molecular and functional characteristics of maltose and maltotriose transporters in Saccharomyces yeasts, particularly given the variability in this trait observed across strains. Overall, the review is quite clear, with a well-written introduction highlighting the importance of sugar transport and maltose/maltotriose in fermentations and brewing. Several sections would gain from further clarification (see below), and while some parts are mostly descriptive, the review will be of interest for researchers specializing in sugar transport and in the molecular mechanisms of brewing.

Suggestions for clarifications and/or minor issues:

- Given the importance of TMH7 and TMH11 (line 97), it would be very useful to annotate these helices in Fig 3. This may also help Figs 6, 7, 8.

- It is not clear that Fig 9 is needed at all, as it only displays a classical degradation pathway common to many proteins, and already explained in the text. Citing a specialized review may be a better alternative.

- line 121 indicates that MALx1 proteins share 95% identity, but then AGT1(MAL11) is indicated to have 57% identity with Mal61 at line 187. Can the authors clarify if Agt1 is Mal11 or a Mal11-like gene, as the given numbers sound contradictory and makes it unclear what is the general % identity between these proteins. Can the authors clarify the text or, alternatively, provide an overview of % identities (example: table or heatmap)?

For example, the text in lines 235-237 is clearer.

- The locus specifications in lines 204-209 could be made clearer with a figure (like Fig 4).

- “Table 3 legend” appears at an unexpected location (lines 104-108)?

- species italics missing in lines 162-163, 165, 220-222, 249, 293, 312-316, 320, 325-329, 332, 337, 353, 369, 535 (not exhaustive). Likewise, some italics is inconsistent for genes, see for example line 389 and line 396.

- line 275: it would be informative to say whether the differences between MTT1 and MTY1 are located at key positions in the protein (such as transmembrane domains or substrate-contacting residues).

- line 425: clarifying “subfunctionalizing escapes from adaptive conflict” would be helpful for readers, in particular what constitutes “adaptive conflict” in this case? Or are the two main contrasting hypotheses neofunctionalization (i.e. transporter accepts a novel substrate) vs subfunctionalization (i.e. the ancestral transporter accepts many substrates, and each paralog “specializes” in a subset)?

- Table 1: “L69 y L70” -> “L69 & L70”

- line 445: the start of the sentence is missing, assumed to be a repetition of the title?

- line 531-532: the residues should be written as elsewhere in the text (i.e. Q156, L164, etc) rather than differently (156Q, 164L, etc), for consistency across the paper.

- line 602: check gene name format?

Author Response

In this manuscript, the authors review the molecular and functional characteristics of maltose and maltotriose transporters in Saccharomyces yeasts, particularly given the variability in this trait observed across strains. Overall, the review is quite clear, with a well-written introduction highlighting the importance of sugar transport and maltotriose in fermentations and brewing. Several sections would gain from further clarification (see below), and while some parts are mostly descriptive, the review will be of interest for researchers specializing in sugar transport and in the molecular mechanisms of brewing.

We thank the reviewers for their thoughtful feedback, have addressed the comments and suggestions as thoroughly as they were requested.

It is important to mention that according to reviewer 1, we should add one figure. It is shown at Page 7, lines 229-232. For these reason figures number were modified. Since now, figure 5 has been incorporated and original figure 9 was deleted.

  Suggestions for clarifications and/or minor issues:

- Given the importance of TMH7 and TMH11 (line 97), it would be very useful to annotate these helices in Fig 3. This may also help Figs 6, 7, 8.

Please note that the figures number has changed. Older 6, 7 and 8, now they are 7, 8 and 9.

Helices mentioned above have been highlighted. Modifications can be seen at Figure 3 (Page 4, lines 114-117); Figure 7 (Page 17; lines 603-605); Figure 8 (Page 18, lines 627-629).

As for Figure 9, it depicts the intracellular region of the transporter, where the transmembrane helices are either not visible or only partially shown; therefore, we consider it unnecessary to annotate them in that figure.

- It is not clear that Fig 9 (original) is needed at all, as it only displays a classical degradation pathway common to many proteins and already explained in the text. Citing a specialized review may be a better alternative.

According to the suggestion, Figure 9 was removed. In its place, we now cite a specialized review that describes the degradation pathway mentioned in the text, as recommended. Page 19 Line 662.

- line 121 indicates that MALx1 proteins share 95% identity, but then AGT1(MAL11) is indicated to have 57% identity with Mal61 at line 187. Can the authors clarify if Agt1 is Mal11 or a Mal11-like gene, as the given numbers sound contradictory and makes it unclear what is the general % identity between these proteins. Can the authors clarify the text or, alternatively, provide an overview of % identities (example: table or heatmap)?

The reviewer is right in pointing out the potential confusion. Since AGT1 is an allele of MAL11 and both genes occupy the same locus (MAL1), many authors and databases refer to AGT1 as MAL11. However, it is important to distinguish between the originally described MAL11 and its allelic variant AGT1, as they encode proteins with significantly different sequences and characteristics. We have added a clarifying paragraph in the revised manuscript to clearly explain this distinction (Page 6, lines 192-200). We believe that with this revision, the percentage identity differences are now much clearer.

For example, the text in lines 235-237 is clearer.

- The locus specifications in lines 204-209 could be made clearer with a figure (like Fig 4).

We agree with the suggestion and have added a figure illustrating the canonical structure of the MAL1 locus harboring the AGT1 gene (Page 7, lines 229-232). Additionally, we included a brief description of the regulatory elements in the AGT1 promoter region (Page 7, lines 225-228).

- “Table 3 legend” appears at an unexpected location (lines 104-108)?

Table 3 does not exist through manuscript. This mistake could be due to formatting issue during the submission process.

- species italics missing in lines 162-163, 165, 220-222, 249, 293, 312-316, 320, 325-329, 332, 337, 353, 369, 535 (not exhaustive). Likewise, some italics is inconsistent for genes, see for example line 389 and line 396.

This also appears to be a formatting issue possibly introduced during the file processing stage. In our submitted version, all species and gene names were carefully formatted in italics, and we verified this prior to submission. We hope this issue will not persist in the revised version.

- line 275: it would be informative to say whether the differences between MTT1 and MTY1 are located at key positions in the protein (such as transmembrane domains or substrate-contacting residues).

Amino acid differences between MTT1 and MTY1 are indicated at page 8, lines 301-305. Above residues are distributed across cytoplasmic and extracellular loops, as well as within transmembrane helices (TMHs), including TMH5 and TMH6.

- line 425: clarifying “subfunctionalizing escapes from adaptive conflict” would be helpful for readers, in particular what constitutes “adaptive conflict” in this case? Or are the two main contrasting hypotheses neofunctionalization (i.e. transporter accepts a novel substrate) vs subfunctionalization (i.e. the ancestral transporter accepts many substrates, and each paralog “specializes” in a subset)?

To improve the information, the original phrase “subfunctionalizing escapes from adaptive conflict” was removed, and the explanation reformulated according to the evolutionary mechanism proposed by Crandall et al. 2024 [82]. This change aims to retain the conceptual content of the original citation while improving accessibility for a broader readership.

- Table 1: “L69 y L70” -> “L69 & L70”

The text has been corrected to read “L69 & L70” in Table 1.

- line 445: the start of the sentence is missing, assumed to be a repetition of the title?

The sentence was changed from “Exist a very high number of polymorphisms…” by “There is a very high number of polymorphisms among all permeases of this type” (Page 13, line 476)

- line 531-532: the residues should be written as elsewhere in the text (i.e. Q156, L164, etc) rather than differently (156Q, 164L, etc), for consistency across the paper.

Annotation residues were corrected as suggested (Page 15, lines 563-564).

- line 602: check gene name format?

The enzymes were initially referred to as “npi1/rsp5” and “npi2/doa4” based on the terminology used in the cited reference [104]. However, after reviewing more recent literature, we adopted the commonly accepted nomenclature “ubiquitin ligase Rsp5” and “ubiquitin hydrolase Doa4,” and have updated the text accordingly (Page 18, lines 633-634).

Reviewer 2 Report

Comments and Suggestions for Authors

I don’t think that it is suitable to use first person to write scientific articles, especially review articles. So, no ‘we did’, but ‘it was done’. In my opinion, this aspect has to be corrected throughout the manuscript.

The arrangement of the article is also untypical. You have divided this manuscript in sections, and in some section have put just one subsection (4; 4.1; 5; 5.1…). if you have one subsection, is there really need for it? Couldn’t it be continuous text?

During the manuscript, the authors always describe the ability of the yeast to assimilate larger pool of sugar as an advantage. This is generally the case, but due to the current trends, perhaps some paragraph or two could mention advantage of the lower attenuation of some of the yeast species for the production of low-alcoholic beverages?

  1. 39-41 I think that it is important to note, that the baker’s yeast and distillers yeast is S. cerevisiae.
  2. 104-108 Table 3 is mentioned. Firstly, there is no table. Secondly, why ‘3’ if it should be (if it were there) ‘1’.

Figure 3 is not mentioned in the text.

  1. 162-165 Italics for Latin names.
  2. 220-222 Italics. Also, L. 249. Correct throughout the manuscript.
  3. 445 ‘Exist’? I suppose it should be ‘there exist’ or something similar.
  4. 470-472 It is one of this, but is it substantial, compared to others, or not?
  5. 683 I think, ‘conclusion’ not ‘concluding remarks’

Author Response

I don’t think that it is suitable to use first person to write scientific articles, especially review articles. So, no ‘we did’, but ‘it was done’. In my opinion, this aspect has to be corrected throughout the manuscript.

According to the reviewer’s suggestions, all paragraphs where we use of the first person were modified. As can be seen on pages 1, lines 18-25; p 2, line 53; p 5, lines 149-150; p 10, line 369; p 12, line 466; p 13, line 477; p 15, line 561 and p 21, lines 706-707.  

The arrangement of the article is also untypical. You have divided this manuscript in sections, and in some sections have put just one subsection (4; 4.1; 5; 5.1…). if you have one subsection, is there really need for it? Couldn’t it be continuous text?

Structure of the paper was organized following the IJMS formatting style. However, it was adjusted as suggested for reviewer 2. Pages 4-9

During the manuscript, the authors always describe the ability of the yeast to assimilate larger pool of sugar as an advantage. This is generally the case, but due to the current trends, perhaps some paragraph or two could mention advantage of the lower attenuation of some of the yeast species for the production of low-alcoholic beverages?

Now we have added a paragraph acknowledging that, although the complete assimilation of sugars is generally considered advantageous, certain brewing applications such as the production of low-alcoholic beverages may benefit from yeast strains with lower attenuation. This addition aims to reflect current trends in the industry, as suggested (Page 2, lines 60-63).

39-41 I think that it is important to note that the baker’s yeast and distillers’ yeast is S. cerevisiae.

We added that S. cerevisiae is also used in the baker and distillers’ industry (Page 1, line 40).

104-108 Table 3 is mentioned. Firstly, there is no table. Secondly, why ‘3’ if it should be (if it were there) ‘1’.

Figure 3 is not mentioned in the text.

162-165 Italics for Latin names.

220-222 Italics. Also, L. 249. Correct throughout the manuscript.

These issues may have resulted from minor formatting or conversion problems during the processing of the manuscript for review. Our original version did not include any reference to “Table 3,” as the manuscript contains only one table. It is possible that the line referencing Figure 3 was inadvertently removed or displaced. Additionally, all species and gene names were properly italicized in our submitted draft, and we thoroughly verified this before submission. We trust these issues will not reappear in the revised version.

445 ‘Exist’? I suppose it should be ‘there exist’ or something similar.

The sentence was changed from “Exist a very high number of polymorphisms…” by “There is a very high number of polymorphisms among all permeases of this type” (Page 13, line 476)

470-472 It is one of this, but is it substantial, compared to others, or not?

The lines were rewritten to clarify that polymorphisms are determinant in the transport function of these proteins. Moreover, references that remark the importance of polymorphic residues on the permease’s activity were included. However, to our knowledge, there are no reports providing direct comparisons of the relative contribution of these residues versus other influencing factors (Page 14, lines 501-503).

683 I think, ‘conclusion’ not ‘concluding remarks’

We agree with the reviewer and have changed the heading from “Concluding remarks” to “Conclusions” (Page 21, line 712)